# Development and Testing of Performance Scale Application as an Effective Electronic Tool to Enhance Students' Academic Achievements

**Fezile Ozdamli [1],\*** , **Mustafa Ababneh [2],\***, **Damla Karagozlu [3]** and **Aayat Aljarrah [4]**

[1] Management Information Systems, Near East University, Nicosia 99138, Cyprus
[2] Department of Computer Information Systems, Near East University, Nicosia 99138, Cyprus
[3] Department of Management Information Systems, Cyprus International University, Nicosia 99258, TRNC, Turkey
[4] Department of Computer Science, Nasser Vocational Training Centre Jau, Road No. 5712, Jaww P.O. Box 80240, Bahrain
\* Correspondence: fezile.ozdamli@neu.edu.tr (F.O.); 20194017@std.neu.edu.tr (M.A.)

**Abstract:** Performance scale application (PSA) usage in the classroom is underutilized, despite the rapid progress of mobile phone and e-learning technology. Lack of self-learning, evaluation, satisfaction, and inability to choose appropriate specialties influence students' academic achievement in secondary school. The objective of this study is to investigate the development and testing of PSA on students' learning achievement in secondary school. The PSA was developed on the Android mobile operating system using the extra trees regression algorithm to predict student achievement in secondary school. Students in the 11th grade basic specialty were considered. Three specialties were used, namely scientific, literary, and industry. The variables examined include improving evaluation (IME), improving communication (IMC), improving scientific (IMSC), and satisfaction of learning (SOL). The findings demonstrated that the PSA accurately predicted the students' choice of specialty, IMC, IMSC, SOL, personalized learning (L), distance L, mobile L, self L, and specialty L. The findings also indicated a positive and significant effect of the PSA on students' learning achievement. This validates that the extra trees regression is an effective tool for the development of PSA. In conclusion, the PSA has efficiently predicted the choice of specialties and academic achievements of students in secondary schools.

**Keywords:** achievement; machine learning algorithm; performance scale application; prediction; secondary school; specialty

## 1. Introduction

One important aspect of education is student academic achievement [1]. It is regarded as the focal point around which the entire educational system revolves. Students' success or failure is defined by their academic achievement [2]. Student academic achievement is also highlighted as being key teachers' top priority in classrooms as well as the school's highest priority and goal. Academic achievement can also be described as the knowledge acquired that can be evaluated by a teacher through marks and school guidelines during a specified period. In addition to giving students the necessary knowledge and abilities, education also gives them a valuable tool they may use to address difficulties in their academic journey. They must be able to put the knowledge they have acquired in class into practice and apply it to actual circumstances [3]. A learner needs to be taught how to comprehend the teaching materials, especially how to convert a figurative notion into specific nonfigurative data and meaning [4].

Mobile electronic technology plays a role in the learning process by assisting students in both decoding difficult abstract concepts and fostering the configuration of those

concepts in their minds using its identifiable multidimensional tools and technological enablers [5]. Studies have established the benefits of using mobile learning tools to teach various concepts, including how they have boosted student enthusiasm, choice, and making important decisions, while also promoting the learning of diverse subjects in schools [6,7]. The term "mobile learning" refers to the form of instruction that utilizes mobile devices and the services they offer, such as Wireless Application Protocol (WAP), application software, and other technologies [8].

The traditional educational model, which relies on memorization, face-to-face classroom learning, and reliance on the teacher as the center of the learning process and the book as the primary knowledge source, is no longer the most effective model; rather, the rise of technology is credited with the emergence of distance learning tools [9], such as PSA or mobile applications, as a new paradigm. Mobile learning depends on the use of wireless technologies and applications in delivering distance and classroom learning in an effective manner to students [10]. With PSA, students can make correct self-decisions, self-learn outside the classroom, and interact effectively with their teachers and school, as well as connect the learning process to real-world situations.

A performance scale application (PSA) is an electronic mobile device developed to support students' learning process and achievement in secondary school. It enables students to promptly and accurately make decisions. With PSA, students can make correct self-decisions, self-learning outside the classroom, and interact effectively with their teachers and school, as well as connect the learning process to real-world situations. On the other hand, the PSA also provides teachers and schools with opportunities by offering students learning materials including documents, videos, PowerPoints, audios, and chat rooms. Teaching students with visual tools is not a novel concept, but developing mobile applications to guide secondary school students in self-learning and making the right decisions, which improves their academic performance, is a new concept. Despite the improvements and progress in the decades-long use of mobile apps in educational settings, there are still issues with self-learning and students' academic achievement in secondary school. This is because, in Jordan, secondary school students decide on their specialty or class they want to study (whether scientific, literary, or industry) [11]. Their poor decisions have an impact on their academic achievements. Self-learning continues to be challenging among secondary school students, which has a significant impact on students' academic achievement. No study was found that aimed to assist secondary school students in choosing their academic specialty in Jordan. Generally, studies were found to focus on university students [12]. Hence, the primary purpose of this work is to investigate the effect of PSA on students' academic achievement in secondary school. The following research questions guide the objective of this study: RQ1: Is the PSA effectively improving students' academic achievement in specialties? RQ2: In which studied subjects is the performance scale effectively improving students' scores? RQ3: Is the improvement of students' achievement in selected subjects due to chance or use of the proposed application? RQ4: Does the use of data mining techniques improve student performance records at the secondary level?

Based on these questions, this study formulated and tested the following research hypotheses:

**H1:** *There are statistically significant variations in how specialties are represented in the sample of participants.*

**H2:** *The use of the performance scale can improve students' learning achievement by improving their communication.*

**H3:** *The use of the performance scale can improve students' learning achievement by improving their scientific content.*

**H4:** *The use of the performance scale can improve students' learning achievement by improving their learning satisfaction.*

**H5:** *The use of the performance scale can improve students' learning achievement by improving their personalized learning.*

**H6:** *The use of the performance scale can improve students' learning achievement by improving their distance learning.*

**H7:** *The use of the performance scale can improve students' learning achievement by improving their mobile learning.*

**H8:** *The use of the performance scale can improve students' learning achievement by improving their self-learning.*

**H9:** *The use of the performance scale can improve students' learning achievement by improving their specialty learning.*

The significant contributions of this study include supporting effective pedagogical delivery through mobile learning apps. There is no study on PSA focusing on using mobile learning platforms and students' learning achievement in secondary schools in Jordan. Thus, by doing this, this study will contribute to the body of knowledge. The use of machine learning algorithms in the development of PSA with user interface principles aims to assist secondary school students in their self-learning, self-decision, specialization choice, and academic work, including classroom assignments. Doing this can help the student to make the correct choice of specialty, boost their classroom interest and participation, improve their examination outcomes, and improve their entire learning achievement in secondary school. The findings of this study can also assist schools in making appropriate use of the curriculum and maximizing their use of time in impacting knowledge, rather than the inappropriate use of time training students to make decisions or assigning students to the wrong specialty or area of interest, resulting in failures and poor performance.

## 2. Literature Review

### 2.1. Performance Scale Application

The term "performance scale" refers to a scale from which the degree of achievement for any given level of actual performance is determined [13]. A PSA is a form of software created specifically for mobile electronic devices such as Android phones and tablets [14]. Similar services to those accessed on PCs are routinely made available to users through mobile applications [15].

These portable computer devices have been positively portrayed because of the rapid expansion in their use and have become a trend that attracts students' attention. Students have the opportunity to completely express their innovation in a structured environment utilizing mobile electronic technologies that can be visualized [16]. The students can use, upload, and share information with others that is directly related to their learning. The PSA can improve students' responses to learning outcomes.

The PSA is part of an e-learning initiative aimed at revolutionizing the education system, including secondary level, across the globe [17]. All stakeholders, especially students and teachers, play a significant role in the effective integration of e-learning systems and distance learning platforms in secondary education [9]. With the introduction of PSA in secondary education, students and teachers must be willing to use e-learning tools in the classroom. Through e-learning systems and remote learning platforms, such as mobile learning platforms, teachers can virtually reach a larger population of students [18].

As part of their learning process, schools can provide students with learning materials in a variety of ways, including applications, games, videos, PowerPoints, audios, chat rooms, and computer testing systems [19]. On the other hand, students can access, exchange, and locate educational materials to enhance their learning on PSA [20]. Despite the information, there is currently little use of PSA as e-learning systems, particularly in developing nations, and further research is still required in these nations (e.g., Jordan) to encourage their use in secondary schools.

### 2.2. Student's Academic Achievement

Academic achievement is a measure of performance. The degree to which a student, instructor, or school has realized their set educational objectives within a set period is known as academic achievement or academic performance [21]. Academic achievement depends strongly on the outcomes in the classroom, showing how well students have met the learning objectives [22].

A number of studies have focused on developing a device for forecasting student achievement, dropout rates, self-decision, self-learning, classroom performance, and reducing exam cheating rates within e-learning systems [23,24], while others have e-learning applications, such as PSA, to predict student performance and student assessment [25,26]. The majority of the data from these studies was gathered through student information systems. In contrast to earlier research, some studies focused on using machine learning algorithms to assess instructors' performance and help them become more effective [27,28].

Previous research has demonstrated that one of the most crucial methods for achieving academic success is the practice of predicting academic achievement using performance scale metrics to enhance student performance [29–31]. The need to develop new mobile software to direct secondary school students toward the suitable specialty for their scientific aptitude based on their academic achievement is important in boosting student performance in schools.

### 2.3. Choice of Specialty

Most students want to select a specialty they actually enjoy, but they frequently lack the experience in a range of areas to make an informed choice [32]. Many pupils find it challenging to accept this one. It is beneficial to establish a backup specialty that a student would enjoy practicing in addition to acquiring experience in the preferred specialty, particularly if the student's initial choice is in a challenging specialty or area [33]. Through PSA, students' chances of being in the right specialization might be improved [33,34]. Important factors to consider while making the choice include test results, interest in the field, and good recommendations. The most important reasons why secondary school students choose a specialization are improved potential, interest, ease of passage, a well-known area, an award, future opportunities, and admiration [33,35]. These are important factors considered by schools and regulators. The research on students' choices of specialty in secondary schools is scarce. Therefore, more focus is needed to fill this gap in research.

### 2.4. Data Mining Methods

An intelligent educational system is one of the common methods used to mine and analyze information. The use of artificial intelligence and machine learning techniques has made the e-learning data mining process easier, more reliable, and higher in precision [36]. Many studies have built systems to evaluate students and predict their academic performance using automated learning algorithms and have reported high accuracies [9,28,37]. With the use of educational data mining tools, it is possible to process educational data supplied by a variety of platforms, including e-learning, and automated result management systems to gain incredibly important insights into students' performance [38].

A study has developed an algorithm that works to gather the "GPA," or grade point average [39]. This procedure was carried out by examining a variety of circumstances, including educational trends in academic performance, the impact of academic success on groups, and the teaching strategies for students. The dataset was collected from university students, and the algorithm was validated using clustering analysis. Even though this algorithm demonstrated its success, data collection in this case is limited to the students' perspective. Based on historical student academic data, Ref. [38] used machine learning with binary classes to forecast student performance (prediction of grades).

A study identified students who failed to keep up with the programming classes, instead of assessing students' performance using the clustering method [40]. The student profiles were used by the researcher to gather the data and it was discovered that the usage

of K means in the algorithm produced accurate results. Additionally, a different study developed an algorithm to improve students' academic performance by identifying those who had late homework. The researcher obtained the data by grouping the students in a mixed-educational course and using an internet scholar course [41].

The researcher developed a visual analysis technique termed "performance Vis" for use in another research report [42]. Utilizing three key factors, student characteristics, homework completion, and exam question design, the study examined data on students' performance over time. The technique manages the students while enrolled in the course. The information system used by the students to store the data was used in the design of this technique. However, the findings showed the effectiveness of the performance Vis in predicting the students' performance in a course. It would have been preferable if the performance Vis had been developed to correspond with the course while it was still in operation. The benefits will, therefore, be greater if the students can immediately identify their areas of weakness. It could also have aided instructors and administrators in identifying students who are likely to fail in a short period of time.

## 3. Materials and Methods

### 3.1. Participants and Sampling

This study developed, gathered, and analyzed data using a quantitative method. A total of 74 students from the 605-student Irbid Secondary School in Jordan participated in this study. The students ranged in age from 16 to 17. An equal number of participants were considered in terms of gender: 37 male and 37 female students. Students were sampled using simple sampling based on the quantitative method of mobile application user interface design proposed by [43]. These students were divided into three groups: the scientific, literary, and industrial specialties, each with twenty-five students. A questionnaire instrument (Table 1) was used to collect data from the students. The items had response choices on a five-point Likert scale (1 = entirely disagree, 5 = totally agree), which were used to assess the intervention's influence on students. These items include improving evaluation (IME), improving communication (IMC), improving scientific (IMSC), and satisfaction of learning (SOL). Table 1 displays the distribution of the questionnaire variables.

**Table 1.** Questionnaire variables distribution.

| Axe No. | Variables | Symbol | Sub Axe | | No. Items |
|---|---|---|---|---|---|
| 1 | Personal Data | PD | Specialty: Scientific, Literary, Industrial | | 1 |
| 2 | Performance scale (independent variable) | IME | Improving evaluation (* X1, X2, ... , X5) | | 5 |
| 3 | Mobile learning skills (dependent variables) | IMC | Improving communication (* A1, A2, ... , A7) | Personalized learning A1, A2, A3 | 3 |
| | | | | Distance learning A4, A5, A6 | 3 |
| | | | | A7 | 1 |
| | | IMSC | Improving scientific content (* B1, B2, ... , B5) | Self-learning B1, B2, B3 | 3 |
| | | | | B4, B5 | 2 |
| | | SOL | Satisfaction of learning (* C1, C2, ... , C10) | Specialty learning C1, C2, C3, C4 | 4 |
| | | | | Mobile learning D5, D6 | 2 |
| | | | | C7, C8, C9, C10 | 4 |

IME = improving evaluation; IMC = improving communication, IMSC = improving scientific, SOL = satisfaction of learning, PD = personal data. * Letter A1–A7, X1–X5, B1–B2, C1–C10 = Number of items.

### 3.2. Cronbach's reliability

The performance scale's contributing factors took use behavior into consideration. The acceptable Cronbach's reliability coefficient ranges from 0.977 to 0.978. It was approved if the Cronbach's alpha and composite reliability scores were equal to or higher than 0.70 [44,45]. Three criteria were used in the study to assess the discriminant validity: the

variable's index value must be less than 0.80 [46], the average variance extracted rate must be at least 0.5, and the average variance extracted (AVE) square must be more than the inter-construct correlations (IC) components [47]. Furthermore, confirmatory factor loadings were 0.7 and higher.

### 3.3. Research Tool

This study developed a PSA for the Android mobile operating system (Figure 1). The application has two interfaces; one is for the student and the other is for the teacher. Teachers were used to provide the students with instructional materials that were used for feedback and daily tasks based on their performance levels in the class.

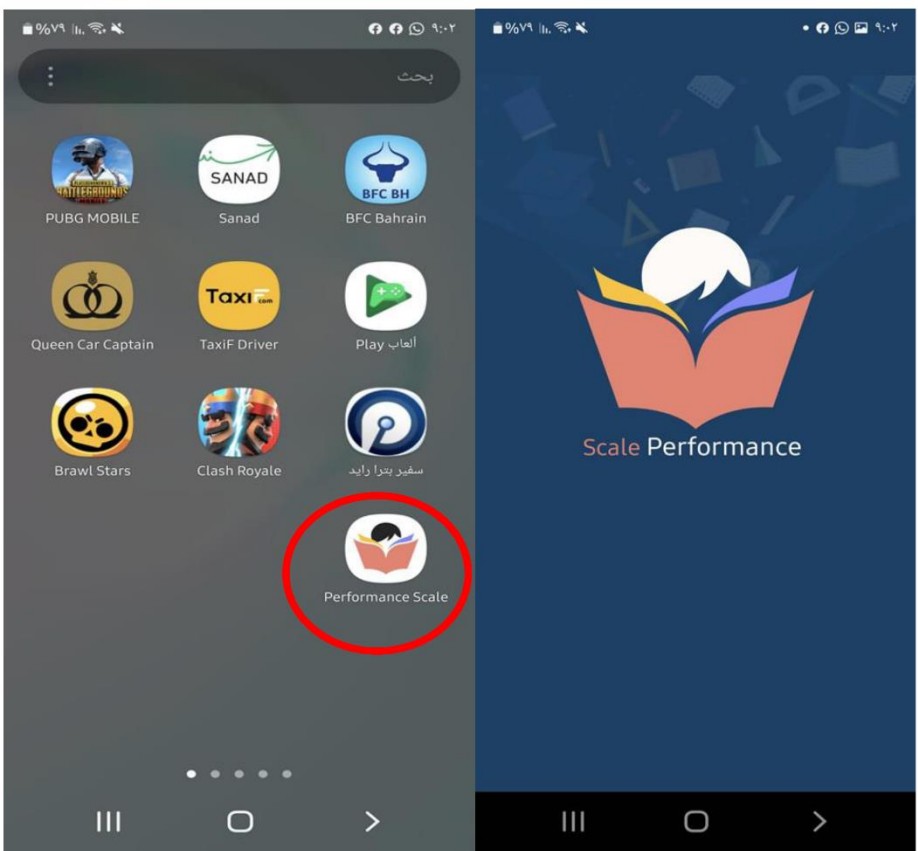

**Figure 1.** The Scale Performance Application.

The students access the academic content offered by the teachers on the PSA. Additionally, they can use the program to turn in homework assignments and privately discuss any concerns they may have with their peers and teacher. Additionally, it enables students to set up meetings with subject-specific advisors to seek support, guidance, and solutions for any problems they could run into when performing their other academic activities. The teacher can also interact with students via messages, answer their questions, and have a conversation with them about their thoughts and concerns using the PSA.

### 3.4. Research Framework

This strategy seeks to anticipate students' secondary school classes before they join the 12th grade, preventing them from being forced to choose a specialty that is not a good fit for their skills based on their previous grades. This is based on a level exam in fundamental courses for every secondary school specialization and an average of the fundamental courses taken in the selected specialty in the 11th grade.

The framework consists of three main components:

The first component is a smart model based on ML algorithms to predict the GPA of the student in the event that the student joins a particular specialty. Many ML algorithms were tested. These include Decision Tree, Extra Trees Regression, Artificial Neural Network, K-Nearest Neighbor, Random Forest, Linear Regression, Bayesian ridge regression, Gradient Boosting, Model Stacking, and Support Vector Machine. Subsequently, the system (PSA) was compared among these models to choose the best model based on efficiency and accuracy.

The second component is the mobile app (PSA) employed to first take students' marks in the basic courses for the specialty that they passed in the 11th grade. Then, it sets an exam level for the specialty chosen by the student, and the result of the exam is a percentage. The application's final output is an AVG output for the student achievement in the 11th grade basic courses and exam results.

The third component is a genetic and fuzzy model, which is used to predict students' achievement and guide them to the best specialty. The model chooses the best result, which appears in the PSA (Mobile App; See Figure 1).

In addition to these, many tasks were included:

Firstly, the average of students' marks in the basic courses for the 11th grade:

This step is taken because, at this level, students have only partially chosen their specialization and still have time to change their minds before they enter the 12th grade. They would not be able to change their majors throughout this grade. Additionally, this plan is crucial for enhancing the system so that the course and environment of study in this class are comparable to those of the 12th grade, including the complexity and content of the courses. This makes it easier to determine whether students have the skills and level of productivity necessary to complete this course effectively. The outcome in this stage is the average.

Secondly, making a level examination for the main courses for each specialty:

This stage is crucial for boosting and enhancing the accuracy of the system operation, as the exam's outcome would reveal the student's performance level. As a result, the outcome of this step is expressed as a percentage, and the average is used to calculate the total of the outcomes of the first and second steps as well. In this case, the average serves as one of the prediction system's inputs.

Thirdly, the other input is the expected GPA:

In this final stage, the genetic algorithm will receive the results from the mobile app, which will include the predicted GPA, average level test scores, and basic course grades. Additionally, the level exam results and the average of the main course grades are sent from the mobile app to the machine learning (ML) genetic algorithm via the full-input API. This is accomplished by writing a Java code because these data are considered inputs for the genetic algorithms. The ultimate output of this proposed system is supplied to display on the mobile app after finishing the genetic algorithm's anticipation of the student's performance regarding their best specialization.

*3.5. The System Development*

3.5.1. Machine Learning Algorithm

Out of ten tested algorithms for prediction accuracy, the extra trees regression algorithm turned out to be the best. Figure 2 displays the result of the accuracy prediction of all tested models. The extra trees regression had an accuracy of 98.92% compared to other algorithms with the three specialties tested, and also provided high precision and high $R^2$ (Figure 3).

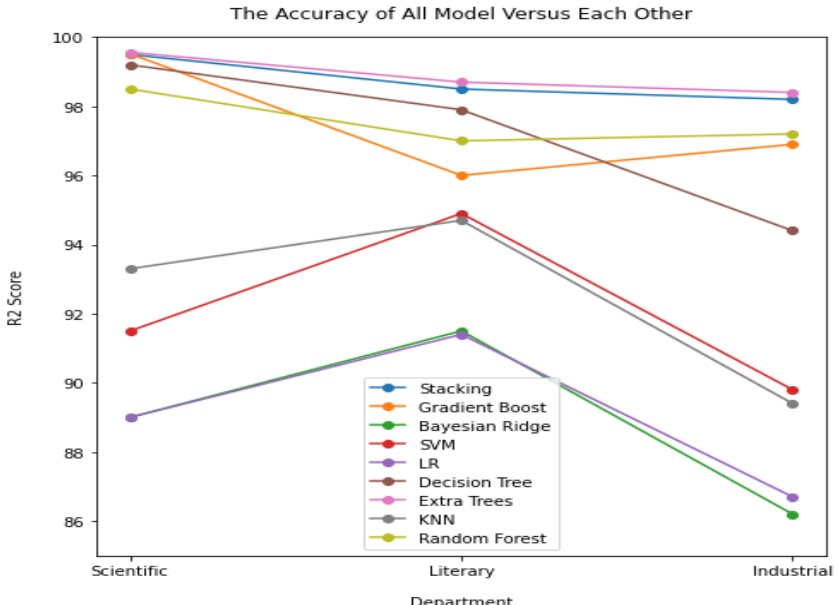

**Figure 2.** Accuracy of all tested models. SVM = Support Vector Machine, LR = Linear Regression, KNN = K-Nearest Neighbor.

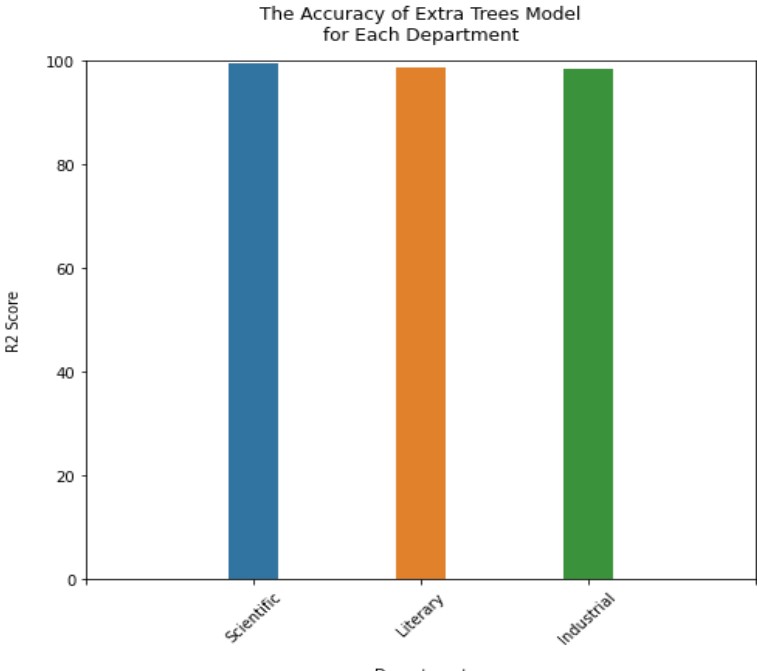

**Figure 3.** Accuracy of Extra Tree Model.

(A) *Supervised learning*

A supervised learning algorithm was employed to spontaneously create an input–output function (a predictor) f (x): M Z to compute estimates of outputs as a function of inputs given a sample (mi, zi)$^N$ of input–output sets (where m i $\in$ M and yi $\in$ Z)[1]. The algorithm searches in the hypothesis space H, which is a suppositionally big but limited set of input–output functions (a subset of the space Z M). Extra trees regression hypothesis spaces are used.

The collection of all samples with finite sizes is represented by

$$(M \times Z)^* = (M \times Z)^N \tag{1}$$

N = 1
*A*: (M × Z)* → H
from (M × Z)* into the hypothesis space H. For a given $H_1$
sample *ls* ∫ (M × Z)* represents *A(ls)* with the function replaced by the algorithm *D*.

We designate by *D(ls)* the function produced by the algorithm *D* for a particular $H_1$ sample, *ls*(M Z).

The probabilistic validation of the supervised algorithm takes into account *m*: $\ell \to M$ and *z*: $\ell \to Z$, which are two random variables specified by a certain probability space $(\ell, \lambda, \theta)$. Let *PM, Z* represent their combined probability distribution described by $M \times Z$ and let $\ell$: $Z \times Z \to R+$ be a non-negative loss function described by $Z \times Z$, while any *f δ* H is represented by

$$L(f) = X \qquad M \times Z \qquad ℝ(f(m), z)dP \, M,Z \qquad (2)$$

(B)　*Linear predictor*

The extra trees regression was used as a linear predicting function *f (k, i)* to estimate the probability that observation *i* has output *k*, in the subsequent function:

$$f(k,i) = \beta_{0,k} + \beta_{2k} \, x_{2,i} + \ldots + \beta_{M,k} \, x_{M,i}$$

where $\beta_{M,k}$ is a regression constant linked with the *m*th instructive variable and the *k*th output. The regression constant and instructive variables are typically classified into vectors of dimension *M* + 1, such that a more concise version of the predicting function is represented as:

$$f(k,i) = \beta_k \cdot x_i$$

where $\beta_k$ is the set of regression constants linked with output *k*, and a row vector is the group of instructive variables linked with observation *i*.

3.5.2. Extra Trees Algorithm

Extra trees: Function for training extra trees classifier or regression.
This function performs the extra trees developing algorithm (executed in Java).

---

**Algorithm 1:**

---

```
   ##
extraTrees(m, z,
            ntree = 605,
            mtry = if (!is.null(z) && !is.factor(z))
                  max(floor(ncol(x)/3), 1) else floor(sqrt(ncol(x)))),
            nodesize = if (!is.null(z) && !is.factor(z)) 7 else 1,
            numRandomCuts = 1,
            evenCuts = FALSE,
            numThreads = 1,
            quantile = F,
            weights = NULL,
            subsetSizes = NULL,
            subsetGroups = NULL,
            tasks = NULL,
            probOfTaskCuts = mtry/ncol(x),
            numRandomTaskCuts = 1,
            na.action = "stop",
            ...)
```

---

(C)　*Set of independent binary extra tree regression*

The extra trees algorithms are obtained running *K* − 1 independent binary extra trees regressions for each of the *K* potential outputs. In these algorithms, few outputs are selected as the "swing," and the remaining *K* − 1 outputs are then individually regressed on the

swing output. If output *K* (the final output) is selected as the swing, then the following occurs:

$$\ln \frac{\Pr(Y_i=1)}{\Pr(Y_i=K)}$$

$$\ln \frac{\Pr(Y_i=2)}{\Pr(Y_i=K)=\beta_1}$$

$$\ldots . \qquad . \; x_i = \beta_2$$

$$\ln \frac{\Pr(Y_i=K-1)}{\Pr(Y_i=K)=\beta_{K-1}} . \; x_i$$

The common Softmax transform, which is employed in compositional data analysis, is another name for this equational algorithm. For each potential result, we have established a distinct series of regression coefficients.

When both sides were exponentiated and the probabilities were calculated, we obtained:

$$\Pr(Y_i = K)e^{\beta_1 \cdot X_i} = \Pr(Y_i 1)$$
$$\Pr(Y_i = \Pr(Y_i = K)e^{\beta_1 \cdot X_i}$$
$$\Pr(Y_i = K\_1) = \Pr(Y_i = K)e^{\beta_{k-1} \cdot X_i}$$

Given that all *K* of the probabilities must add up to 1, we determine:

$$\Pr(Y_i = K) = 1 - \sum_{k=1}^{K-1} \Pr(Y_i = k) = 1 - \sum_{k=1}^{K-1} \Pr(Y_i = k)e^{\beta_1 \cdot X_i}$$
$$\Rightarrow \Pr(Y_i = K) = 1 + \frac{1}{\sum_{k=1}^{K-1} e^{\beta_k \cdot X_i}}$$

This helped us determine the other probability:

$$\Pr(Y_i = 1) = 1 + \frac{e^{\beta_i \cdot X_i}}{\sum_{k=1}^{K-1} e^{\beta_k \cdot X_i}}$$
$$\Pr(Y_i = 2) = 1 + \frac{e^{\beta_i \cdot X_i}}{\sum_{k=1}^{K-1} e^{\beta_k \cdot X_i}}$$
$$\Pr(Y_i = K\_1) = 1 + \frac{e^{\beta_i \cdot X_i}}{\sum_{k=1}^{K-1} e^{\beta_k \cdot X_i}}$$

where, normally, the sum goes from to *K*:

$$\Pr(Y_i = k) = 1 + \frac{e^{\beta_i \cdot X_i}}{\sum_{k=1}^{K-1} e^{\beta_k \cdot X_i}}$$

where $\beta_k$ is described as 0. The algorithm depends on the presumption of independence of unrelated alternatives as indicated above, which is why we conducted extra trees regressions.

(D)  *Extracting, transforming, and feature selection*

(i)  TF-IDF Feature Extractor

To mine text in the system, the feature vectorization technique known as term frequency-inverse document frequency (TF-IDF) is commonly utilized to indicate the significance of a phrase to a particular document in the corpus. Put a term (t) before a document (d) and the corpus (C) before a document. The frequency of a word is expressed as T F (t, d), where t is the number of times the term occurs in document d, and DF (t, C), where t is the sum of documents that include that term. Basically, the sum of TF and IDF makes up the TF-IDF measured.

$$\textbf{T F IDF} \text{ (t, d, C)} = \textbf{T F} \text{ (t, d)} \cdot \textbf{IDF} \text{ (t, C).}$$

where |C| represents the corpus's overall document count. Because a logarithm is employed, a term's IDF value becomes 0 if it occurs in every document. In order to prevent division by zero for terms outside of the corpus, a smoothing term is used.

(ii)  VectorSlicer

A transformer called VectorSlicer was used to accept a feature vector as input and produces a new feature vector containing a sub-collection of the prime features. A vector column's features were extracted using it. The vector column with the supplied index

numbers was input into the VectorSlicer, which subsequently produces a new vector column with the values chosen using those index numbers. AttributeGroup implemented matches on an attribute's name region.

(iii)  Locality-Sensitive Hashing

A significant class of hashing methods known as locality-sensitive hashing (LSH) is often employed in large datasets for outlier recognition, estimate closest neighbor search, and clustering. The basic idea behind LSH is to hash data points into buckets using a family of functions (referred to as "LSH families"), with the aim of making sure that data points that were close to one another are most probable in the similar bucket, whereas data points that were far apart are most probable in different buckets.

An LSH family is a family of functions L that satisfies the following criteria in the metric space (M, d), where M is a cluster and d is a distance function of M.

$\forall$s, t $\in$ M,

d(s, t) $\leq$ r1 $\Rightarrow$ P u(L(s) = L(t)) $\geq$ s1

d(s, t) $\geq$ r2 $\Rightarrow$ P u(L(s) = L(t)) $\leq$ s2

It is known as the (u1, u2, s1, s2)-sensitive LSH family.

### 3.6. Ethical Committee Approval for Individual Security

This is one of the most crucial procedures that researchers must follow while exchanging and gathering data; they must safeguard the data's confidentiality and utilize the data only for the intended purpose [48,49]. An application was submitted to the university's ethical committee in order to obtain approval. The official approval was issued on 16 November 2021.

### 3.7. Data Analysis

The SPSS version 26 program was used to analyze the quantitative data. First, a paired sample t-test was used to determine the mean differences. Second, two tests from each branch were analyzed using Pearson correlation. The difference in averages between groups in each course was then examined using the independent sample *t*-test. AMOS was used to test theories about correlations between intricate variables. The Kaiser–Meyer–Olkin (KMO) and Bartlett's Test were used, which were used to evaluate how well the components explain one another in terms of partial correlation between the variables. If the statistical probability *p*-value for Levene's equality of variance test was higher than 0.05, it meant that the groups had the same variance. The t-test results were then displayed using the "Equal variances assumed" row. However, "Equal variances not assumed" was used for the report if Levene's equality of variance test result was significant. The level of significance was fixed at 0.05.

## 4. Results

### 4.1. Performance Scale Application and its Features

The PSA has two interfaces, a teacher interface and a student interface, which the users can choose from at the start of the system. If the code is available or has already been registered as a user in the database, the user enters it, and the system opens. These functions are based on user interface (UI) functionality.

The PSA development started with the UI, since a strong UI can help users (students, teachers, and schools, with a primary focus on students) maintain continual focus on a selected and targeted object or subject, since any electronic mobile app is categorized based on its purpose and content. The following are the contents of the PSA UI functionalities.

The first component is connectivity, which is the capacity to swiftly and conveniently obtain information. Additionally, it makes it possible for the PSA to be easily stopped, restarted, and resumed. The second component, which consists of three qualities, is ease of use. The application's first feature is to utilize a simple design layout (content, menu, etc.) to ease memory burden. This makes the application easy to recall and encourages the students to pay closer attention. The second feature is brief and concise information so

that the user can read it quickly. The third attribute is divided attention, and since users of electronic mobile devices often need to carry out more than one task at a time, the UI must not be overly complex.

The third component is direction, which refers to the PSA's capability to guide the user through a step-by-step procedure, menu, or choice that necessitates retrieving associated information from the system. The PSA's main strength was its emphasis on the students' ability to make choices. The fourth component provides information to the PSA. It is one of the most useful aspects of UI and a critical prerequisite for transmitting basic and vital information. In fact, if the information is paired with the right configuration, it can be communicated widely. The interaction between students and the PSA, or interactivity, is the fifth component. This element possesses a few pertinent features. To avoid making the landing page overcrowded and losing the interactivity factor, how many options there are on the landing page was taken into account.

Usability is the sixth component. A user-friendly UI was developed in order to achieve a positive and reliable student experience for effective learning. Comprehensiveness or richness is the eighth component. The program must create a complete set of components that enables content transformation. This gives the users (students, teachers, and schools, with a primary focus on students) the chance to control the program in accordance with their own skills and knowledge. This is the second key strength of the PSA's contribution to this development. The eighth component is consistency. Consistency in interactions is a sign of a successful UI. Consistency will assist users to build their opinions and interpretations of the app and will open up opportunity for improvement. Personalization is the tenth component. Experienced students have a strong desire for a sense of control over the UI and for the UI to react to their activities. In order to plan and carry out their learning activities, they will have control in the form of preference. Moreover, the PSA has three (3) types of screen.

➢ First Screen:

Figure 4 shows the first screen in the PSA, the login and sign-up screen. In addition to many other pieces of information, a new student registers as a new user by entering their name, gender, type of study (public or private), and the school's specialization (department). The application gives students a code number to enter into the system after the user is finished.

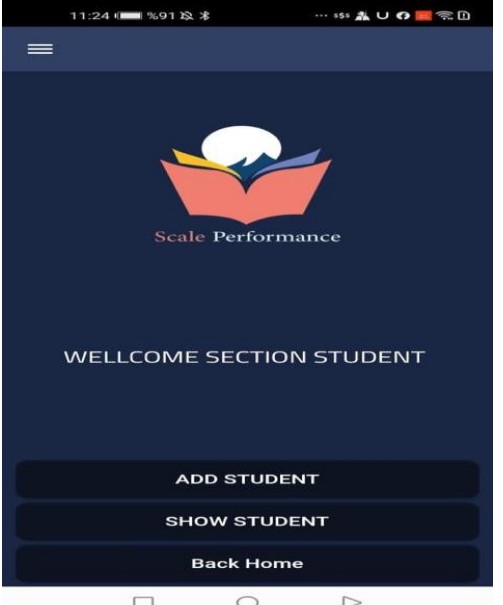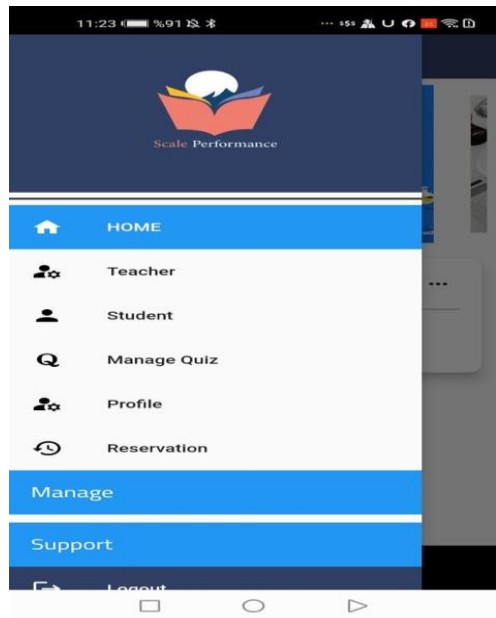

**Figure 4.** *Cont.*

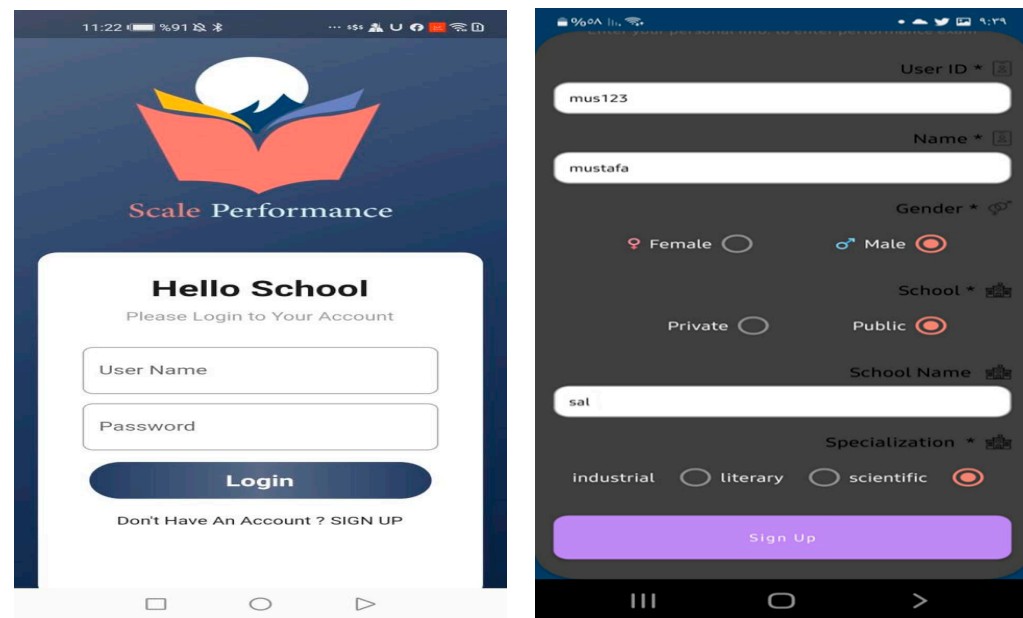

**Figure 4.** The first screen, login and sign-up screen, in the performance scale application.

The application grants access to a screen that displays the prior courses' grades from grade 7 through grade 11 if the applicant selects the study as a private type of choice (Figure 5a). Students are then directed to a screen where they can enter their grades for the foundational courses for the 11th grade specialty they choose (Figure 5b). The app then directs students to level exams created by the ministry of education and by subject-matter experts.

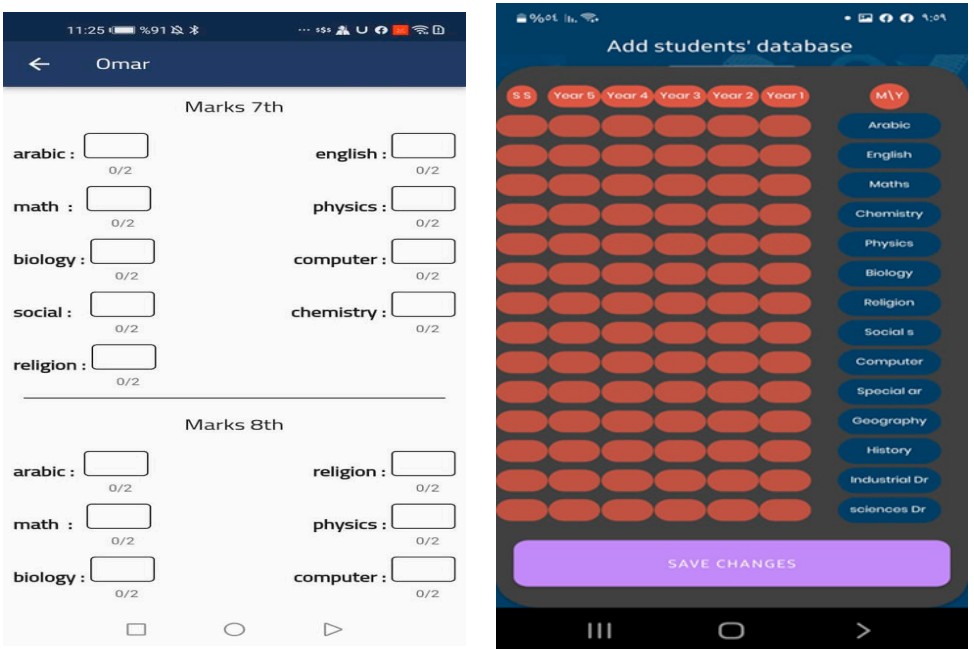

(**a**) Prior grades page of a student    (**b**) Grades of foundational courses

**Figure 5.** The grade screen from 7th to 11th grade.

Figure 6 displays the specialization screen. A screen containing the names of the three specializations (Scientific, Literary, and Industrial; Figure 6a) includes courses in chemistry, physics, biology, and math (Figure 6b).

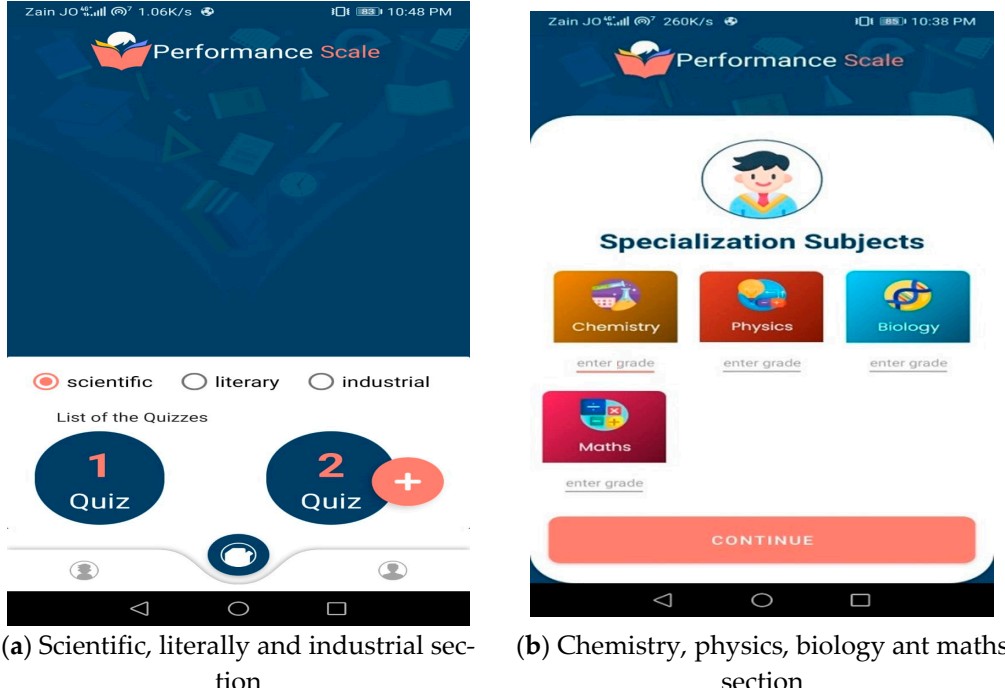

(**a**) Scientific, literally and industrial section

(**b**) Chemistry, physics, biology ant maths section

**Figure 6.** Specialization screen.

The application will direct the user to an empty screen so that the basic subjects' marks, which include those for four courses, can be filled in if the student selects the study as general. The student is next directed toward level tests, which are made up of four exams with the same specialties (specialization, Figure 7a) as the fundamental courses, depending on the student's specialty.

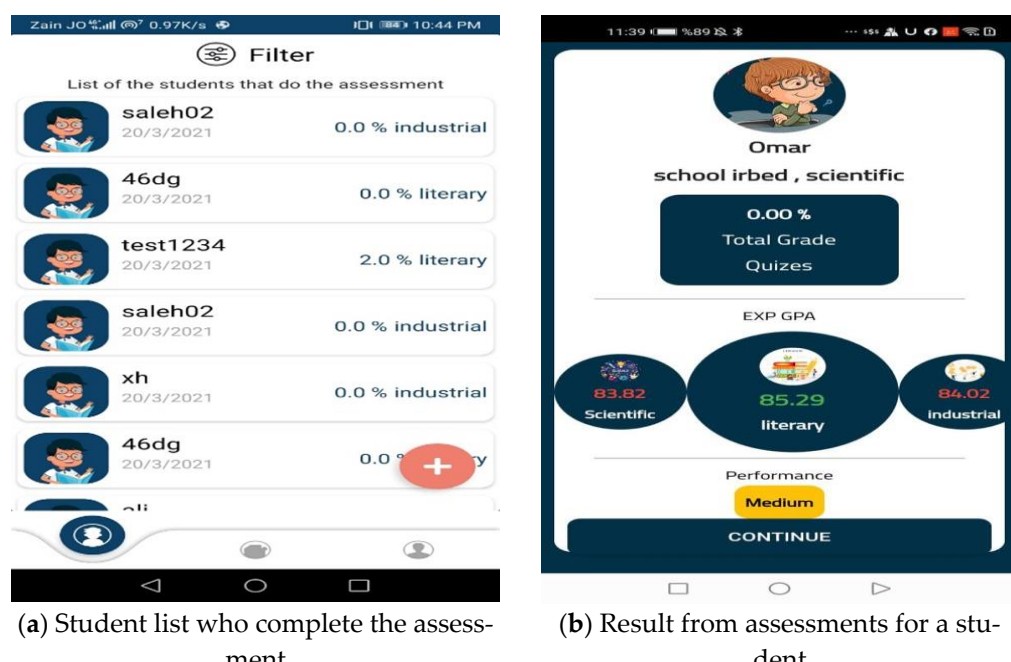

(**a**) Student list who complete the assessment

(**b**) Result from assessments for a student

**Figure 7.** A screen of students' names with their specialization and marks in level exams, the average of the basic subjects, and the students' performance prediction.

➢ Second Screen:

Figure 8 displays the exam control screen, which allows the user to totally control the level tests, especially by adjusting their numbers in response to changes in the courses, the number of questions, and the allotted time for each exam.

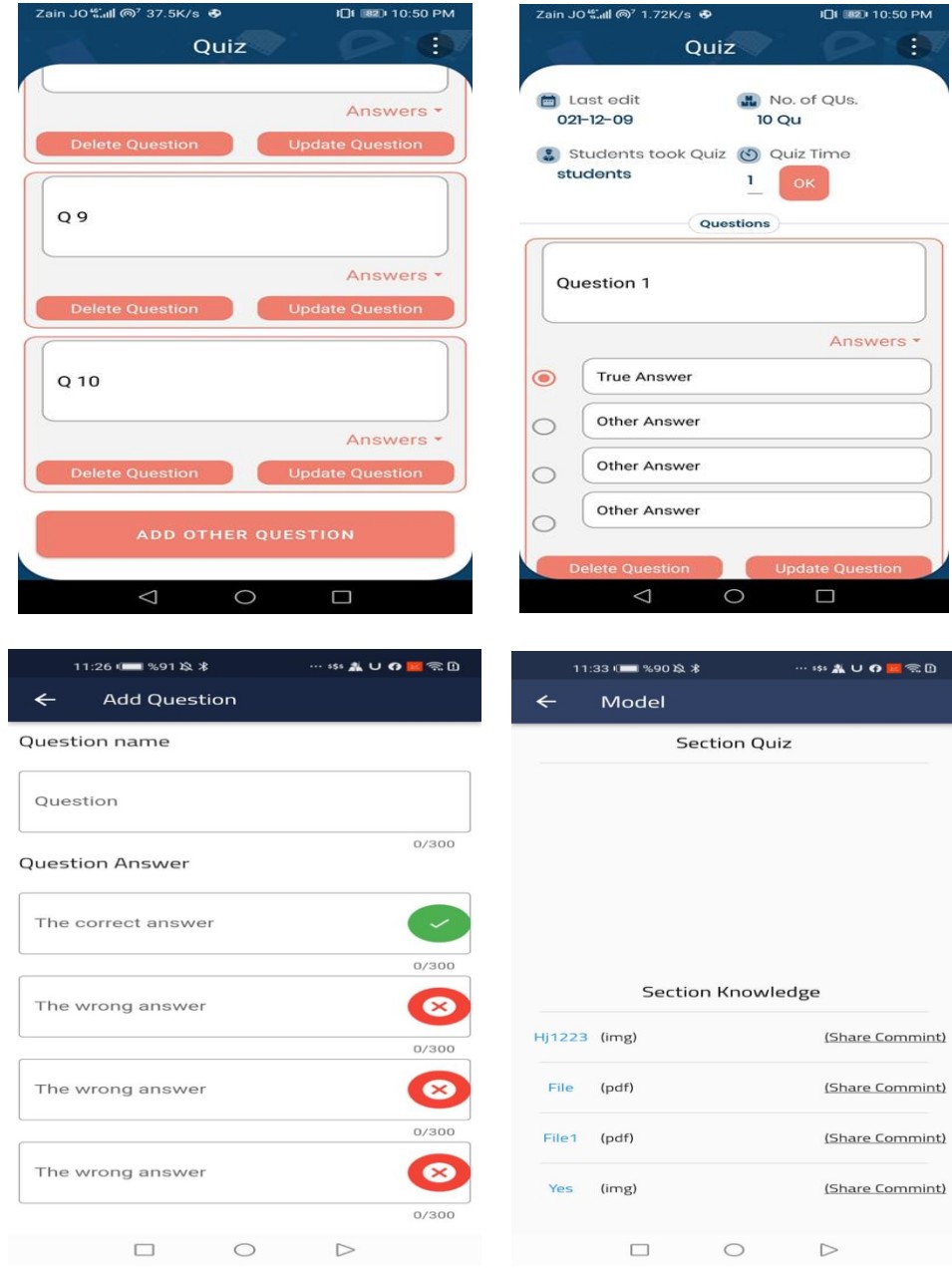

**Figure 8.** Exam control screen.

➢ Third Screen:

Figure 9 displays the screen of the upload. The PSA allows students to upload information and profiles. The student information and grades from grades 7 through 11 can be manually entered into the system. They can upload any quiz information or even share educational links.

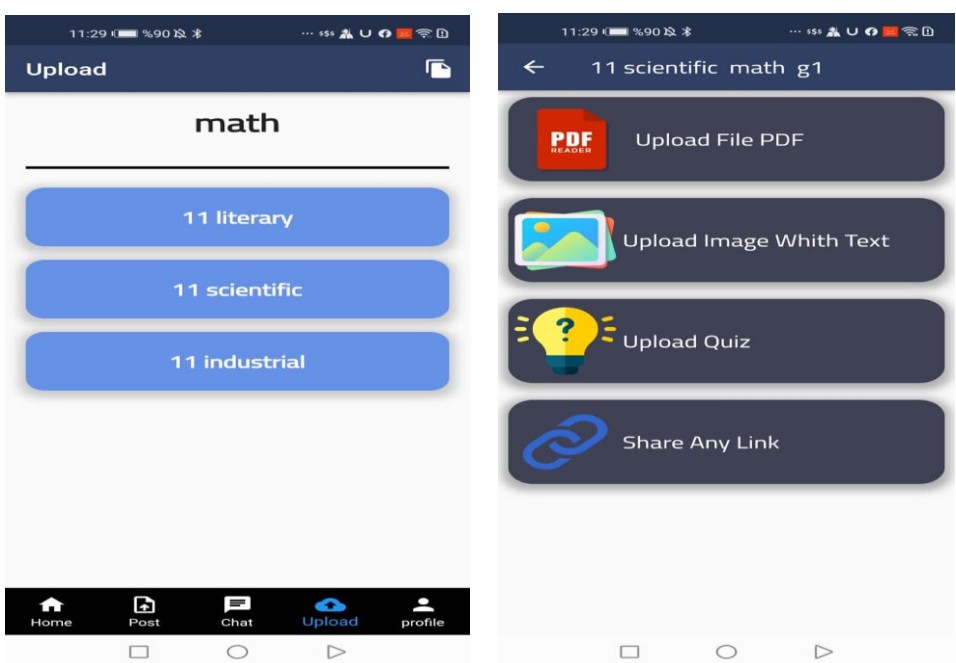

**Figure 9.** The screen of upload.

### 4.2. PSA Response Times

Figure 10 shows the percentiles and mean of response times for a request in the PSA. The 99.9th, 99th, and 95th percentiles were used to determine how good or bad the outliers appeared for the PSA function. These percentiles showed the response time thresholds of the PSA at 99.9 percent, 99 percent, or 95 percent of a request fulfilling in a quicker manner than that of a specific other threshold. When the 95th percentile response time is 2 s, then 95 out of 100 requests will be completed in less than 2 s, while 9 out of 100 requests will take at least 2 s to complete. The grey bars in Figure 10 signify a request, and the height of the bar indicates how long the request took to function. The majority of queries are processed rather quickly, but occasionally, with other outliers, took substantially longer. This may be caused by a number of factors, including loss of a network pack, a junk collection lag, a page error, and mechanical vibrations in the server store.

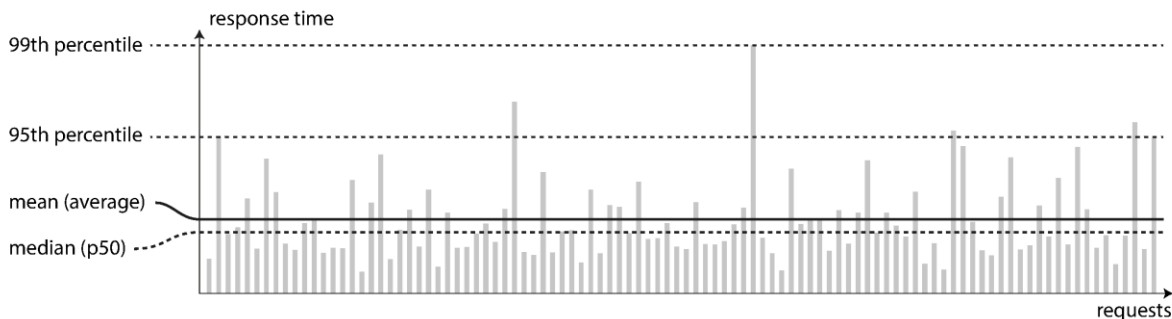

**Figure 10.** The percentiles and mean of response times for a request in the PSA.

### 4.3. PSA Backend Calls

High percentiles develop as particularly crucial when backend activity or response are used repeatedly to fulfill one end-user request. This request still waits for the lengthiest of the parallel calls to finish even if the calls are made in parallel. All requests can become slow with single slow call (Figure 11). The likelihood of receiving a slow call rises if an end-user request necessitates several backend calls, while it is only a little fraction of backend calls that is slow. As a result, a greater fraction of user requests turns out slow.

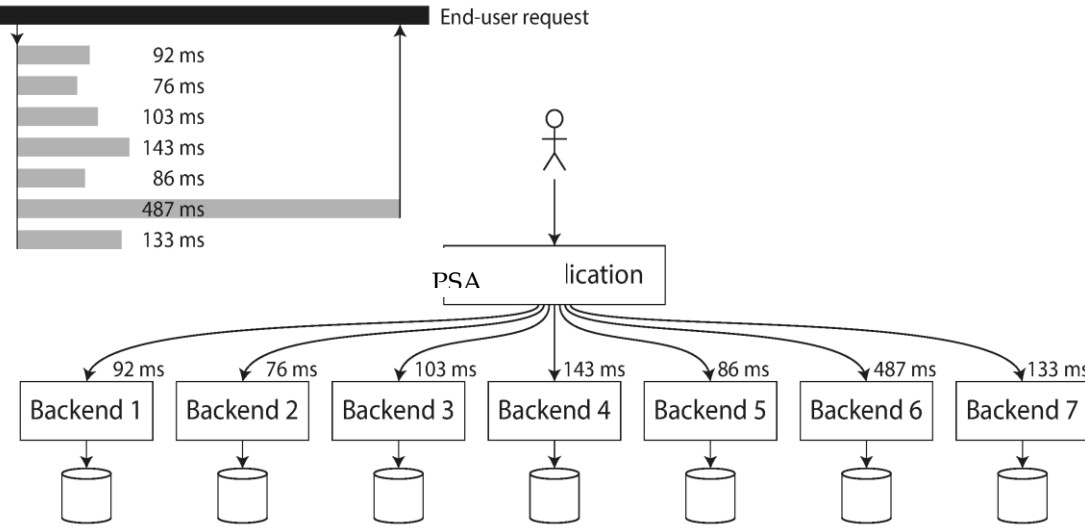

**Figure 11.** PSA backend calls in end-user request.

*4.4. Descriptive Statistics*

As shown in Table 2, a total of 74 students across all specialties, 49% from the scientific field, 41% from the literary field, and 10% from the industrial sector, participated in this study.

**Table 2.** Participant response rate by specialty.

| Specialty | Frequency | Percent |
|-----------|-----------|---------|
| scientific | 36 | 49% |
| literary | 31 | 41% |
| industry | 7 | 10% |
| Total | 74 | 100 |

Table 3 presents the results of KMO and Bartlett's test. The results showed a great KMO measure of sampling adequacy of 0.873. Bartlett's test of sphericity's significant value was good and sufficient for the study to proceed with factor analysis.

**Table 3.** The results of KMO and Bartlett's Test.

| KMO and Bartlett's Test | | |
|---|---|---|
| Kaiser–Meyer–Olkin Measure of Sampling Adequacy. | | 0.873 |
| | Approx. Chi-Square | 2187.205 |
| Bartlett's Test of Sphericity | Df | 351 |
| | Sig. | 0.000 |

The data showed excellent normality distribution values of Variance Inflation Factor (VIF < 10) and tolerance (>0.1) as well as Pearson's product-moment correlation coefficients of less than 0.90 (Tables 4 and 5).

**Table 4.** Results of correlation analysis (dependent variable: SOL).

| Item | Unstandardized Coefficients | | Standardized Coefficients | t | Sig. | Collinearity Statistics | |
|---|---|---|---|---|---|---|---|
| | **B** | **Std. Error** | **Beta** | | | **Tolerance** | **VIF** |
| (Constant) | −0.455 | 0.286 | | −1.589 | 0.122 | | |
| IME | 0.365 | 0.107 | 0.35 | 3.417 | 0.002 | 0.311 | 3.215 |
| IMC | 0.318 | 0.134 | 0.27 | 2.374 | 0.024 | 0.253 | 3.957 |
| IMSC | 0.416 | 0.139 | 0.388 | 2.998 | 0.005 | 0.195 | 5.139 |

Note: IME = improving evaluation; IMC = improving communication; IMSC = improving scientific; SOL = satisfaction of learning; VIF = Variance Inflation Factor.

**Table 5.** Correlation analysis by AMOS.

| Variable | Min | Max | Skew | C.R. | Kurtosis | C.R. |
|---|---|---|---|---|---|---|
| IME | 2.2 | 5 | −0.597 | −2.109 | 0.215 | 0.38 |
| IMC | 2 | 5 | −0.422 | −1.491 | 0.211 | 0.373 |
| IMSC | 2.14 | 5 | −0.595 | −2.103 | 0.381 | 0.674 |
| SOL | 1.8 | 5 | −0.647 | −2.287 | 1.175 | 2.077 |
| Multivariate | | | | | 19.294 | 12.059 |

Note: IME = improving evaluation; IMC = improving communication; IMSC = improving scientific; SOL = satisfaction of learning; C.R. = composite reliability.

### 4.5. Cronbach's Reliability

Table 6 presents the results of the reliability analysis. All the variables meet the criteria of the Cronbach alpha coefficient ranging between 0.977 and 0.978. The item-to-total correlation values for all 27 items were found to have high values beyond the allowed limit. In addition, the coefficient alpha values also demonstrated improved reliability, and the factor loadings of the 27 items were also improved ($p > 0.3$). The findings of this investigation support the high degree of reliability, suitability, and acceptability of the instruments and subsequent data analysis using inferential statistics to test the research hypotheses.

**Table 6.** Results of reliability analysis.

| Cod | Item | Item-Total Correlation | Cronbach's Alpha if Item Deleted | Factor Loadings | Cronbach's Alpha Analysis |
|---|---|---|---|---|---|
| Improving evaluation | | | | | 0.909 |
| | X1* | 0.680 | 0.978 | 0.685 | |
| | X2* | 0.810 | 0.977 | 0.869 | |
| IME | X3* | 0.656 | 0.978 | 0.743 | |
| | X4* | 0.660 | 0.978 | 0.789 | |
| | X5* | 0.704 | 0.978 | 0.852 | |
| Improving communication | | | | | 0.929 |
| | A1* | 0.693 | 0.978 | 0.797 | |
| | A2* | 0.721 | 0.978 | 0.837 | |
| | A3* | 0.744 | 0.978 | 0.824 | |
| IMC | A4* | 0.733 | 0.978 | 0.795 | |
| | A5* | 0.782 | 0.978 | 0.879 | |
| | A6* | 0.737 | 0.978 | 0.780 | |
| | A7* | 0.822 | 0.977 | 0.837 | |
| Improving scientific content | | | | | 0.941 |
| | B1* | 0.778 | 0.978 | 0.858 | |
| | B2* | 0.771 | 0.978 | 0.843 | |
| IMSC | B3 | 0.677 | 0.978 | 0.791 | |
| | B4 | 0.688 | 0.978 | 0.688 | |
| | B5 | 0.768 | 0.978 | 0.826 | |

**Table 6.** *Cont.*

| Cod | Item | Item-Total Correlation | Cronbach's Alpha if Item Deleted | Factor Loadings | Cronbach's Alpha Analysis |
|---|---|---|---|---|---|
| Satisfaction of learning | | | | | 0.954 |
| | C1* | 0.727 | 0.978 | 0.813 | |
| | C2* | 0.765 | 0.978 | 0.840 | |
| | C3* | 0.738 | 0.978 | 0.854 | |
| | C4* | 0.837 | 0.977 | 0.798 | |
| SOL | C5* | 0.833 | 0.977 | 0.862 | |
| | C6* | 0.884 | 0.977 | 0.834 | |
| | C7* | 0.731 | 0.978 | 0.851 | |
| | C8* | 0.753 | 0.978 | 0.774 | |
| | C9* | 0.725 | 0.978 | 0.866 | |
| | C10* | 0.732 | 0.978 | 0.764 | |
| Total | | | | | 0.978 |

Note: IME = improving evaluation; IMC = improving communication; IMSC = improving scientific; SOL = satisfaction of learning. * Letter A1–A7, X1–X5, B1–B2, C1–C10 = Number of items.

To study the rest of the hypotheses, partial least squares SmartPLS 3 software(Version 4.0.8.3, PLS-SEM Academy, Alabama, USA) was used as a key statistical tool to evaluate the outcomes based on confirmatory factor analysis (CFA). Table 7 shows the results of the dependent variables and assessment of the effects described in Figure 6. The constructs, items, and CFA yield factor loading of 0.5 or above were found, which is acceptable [50]. Convergent validity was demonstrated as the average variance extracted (AVE) of the construct's items was above 0.50, which is a highly acceptable value [51,52]. Our analysis indicated the following AVE scores: 0.63 (IME), 0.68 (IMC), 0.65 (IMSC), and 0.68 (SOL). The suitability indexes confirm the model specificity for CR and CA, which meet all requirements, while Cronbach's alpha values varied from 0.847 to 0.948. All were above the 0.70 acceptable value.

**Table 7.** Overall validity and reliability.

| | Fornell–Larcker Criterion | | | | Construct Reliability and Validity | | | |
|---|---|---|---|---|---|---|---|---|
| Item | IME | IMC | IMSC | SOL | CA | RA | CR | AVE |
| IME | 0.791 | | | | 0.847 | 0.853 | 0.892 | 0.625 |
| IMC | 0.849 | 0.822 | | | 0.920 | 0.921 | 0.936 | 0.675 |
| IMSC | 0.830 | 0.846 | 0.804 | | 0.861 | 0.865 | 0.901 | 0.646 |
| SOL | 0.834 | 0.836 | 0.897 | 0.826 | 0.948 | 0.955 | 0.956 | 0.683 |

Note: IME = improving evaluation; IMC = improving communication; IMSC = improving scientific; SOL = satisfaction of learning; AVE = average variance extracted; CA = Cronbach's alpha; CR = composite reliability; RA = reliability analysis.

*4.6. Testsing of Hypotheses*

Multivariate regression is a method for estimating a single regression model with many outcome variables. When a multivariate regression model has more than one predictor variable, the model is a multivariate multiple regression. To test the effects of the specialty on the dependent variables, we employed a multivariate analysis of covariance (MANCOVA; Table 8). This technique was used to account for covariance among the dependent variables. The Box M and Levene's tests (Table 9) of variance homogeneity of the variables included in the model suggest that this assumption was not violated ($p > 0.05$). The MANCOVA findings showed that the specialty had an effect on the dependent variables ($p = 0.005$) as shown in Table 8. Thus, we accept the alternative hypothesis (H1), which states that there are statistically significant variations in how each specialty is represented in the sample of participants.

**Table 8.** Multivariate analysis of covariance: test of overall effect.

| Effect | Wilks' λ | F | df1 | df2 | Sig. | $\eta^2$ | Result |
|---|---|---|---|---|---|---|---|
| Intercept | 0.798 | 5.636 | 3 | 67 | 0.002 | 0.202 | Supported |
| IME | 0.304 | 51.208 | 3 | 67 | 0.000 | 0.696 | Supported |
| Specialty | 0.760 | 3.279 | 6 | 134 | 0.005 | 0.128 | Supported |
| IME - Specialty | 0.803 | 2.585 | 6 | 134 | 0.021 | 0.104 | Supported |

**Table 9.** Box's Test of Equality of Covariance Matrices and Levene's Test of Equality of Error Variances.

| Name of Test | Item | F | df1 | df2 | Sig. |
|---|---|---|---|---|---|
| Levene's Test of Equality of Error Variances | IMC | 1.847 | 2 | 72 | 0.165 |
| | IMSC | 0.456 | 2 | 72 | 0.636 |
| | SOL | 0.027 | 2 | 72 | 0.974 |
| Box's M | 15.127 | 1.102 | 12 | 1257.455 | 0.354 |

Figure 12 and Table 10 demonstrate that IME was significantly and positively associated with IMC ($\beta = 0.849$, t = 6.478, $p < 0.001$). As a result, Hypothesis 2 is validated and accepted, demonstrating that the use of the PSA has improved students' learning achievement by improving their communication. Moreover, IME was found to be significantly and positively associated with IMSC ($\beta = 0.834$, t = 6.707, $p < 0.001$). As a result, Hypothesis 3 is true and accepted, which shows that the use of the PSA has improved students' learning achievement by improving their scientific content. In addition, the findings revealed that IME was found to be significantly and positively associated with SOL ($\beta = 0.830$, t = 6.614, $p < 0.001$). As a result, Hypothesis 4 is true and accepted, which shows that the use of the PSA has improved students' learning achievement by improving their satisfaction of learning. Furthermore, the findings revealed that IME was shown to be significantly and positively associated with personalized L ($\beta = 0.791$, t = 6.491, $p < 0.001$). As a result, Hypothesis 5 is true and accepted and demonstrates that the use of the PSA has improved students' learning achievement by improving their personalized learning. Moving to hypothesis 6, the findings reveal that IME was shown to be significantly and positively associated with Distance L ($\beta = 0.803$, t = 6.154, $p < 0.001$). As a result, Hypothesis 6 is accepted and reveals that the use of the PSA has improved students' learning achievement by improving their distance learning. Similarly, the results demonstrate that IME was shown to be significantly and positively associated with Mobile L ($\beta = 0.721$, t = 6.097, $p < 0.001$). As a result, Hypothesis 7 is accepted, which demonstrates that the use of the PSA has improved students' learning achievement by improving their mobile learning. Likewise, the results demonstrate that IME was shown to be significantly and positively associated with Self L ($\beta = 0.833$, t = 6.547, $p < 0.001$). As a result, Hypothesis 8 is accepted, which demonstrates that the use of the PSA has improved students' learning achievement by improving their self-learning. Finally, the results demonstrate that IME was shown to be significantly and positively connected with IME ($\beta = 0.765$, t = 6.245, $p < 0.001$). As a result, Hypothesis 9 is validated and accepted, which suggests that use of the PSA has improved students' learning achievement by improving their specialty learning (Figure 12 and Table 10).

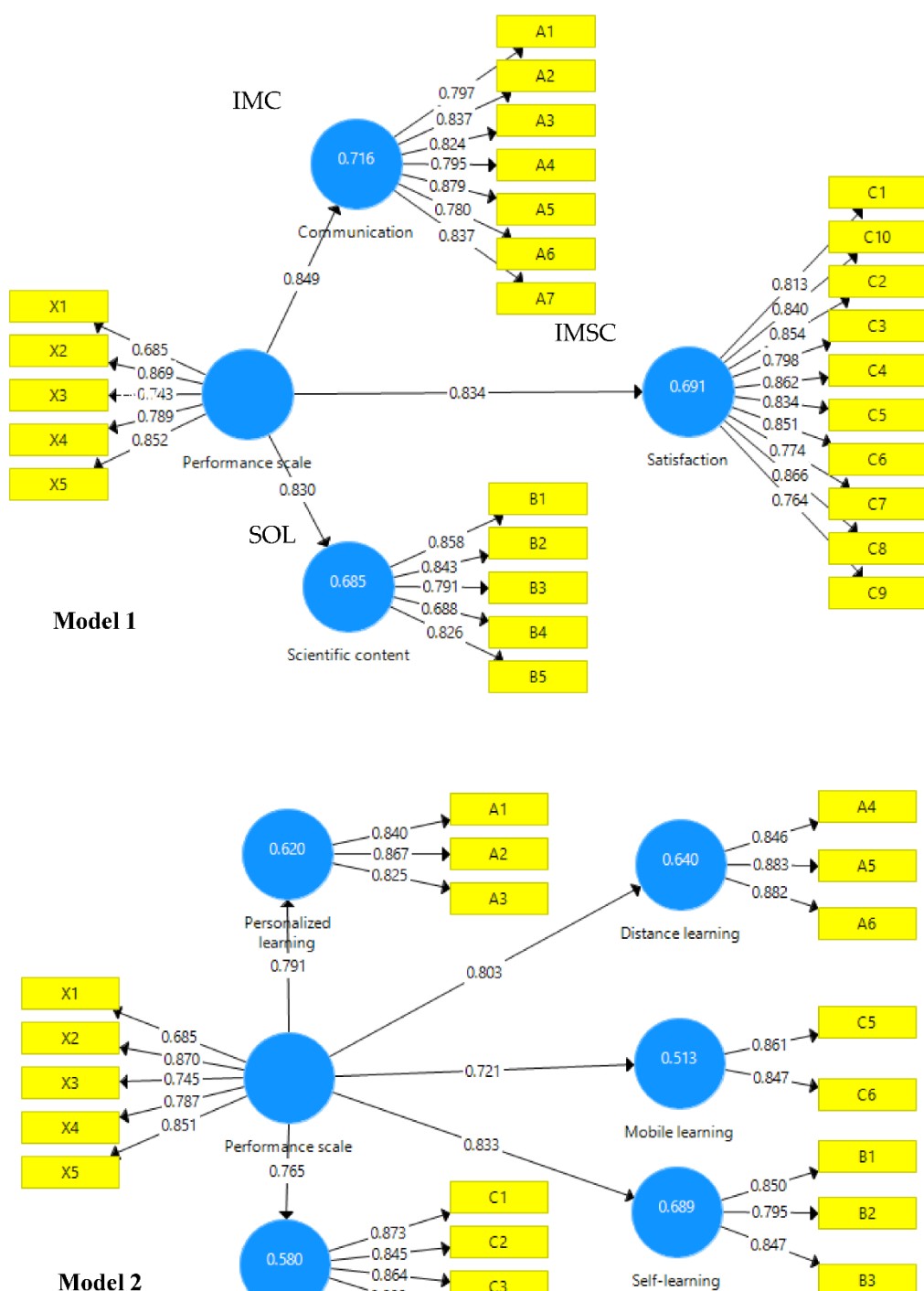

**Figure 12.** Measurement of the dependent variables for the proposed models 1 and 2. Note: IME = improving evaluation; IMC = improving communication; IMSC = improving scientific; SOL = satisfaction of learning.

**Table 10.** Structural models for hypothesis testing results.

| H | Independent | Relationship | Dependent V. | Estimate | S.E. | C.R. | *p* | Result |
|---|---|---|---|---|---|---|---|---|
| H2 | IME | → | IMC | 0.849 | 0.119 | 6.478 | *** | Supported |
| H3 | IME | → | IMSC | 0.834 | 0.138 | 6.707 | *** | Supported |
| H4 | IME | → | SOL | 0.830 | 0.149 | 6.614 | *** | Supported |
| H5 | IME | → | Personalized L. | 0.791 | 0.123 | 6.491 | *** | Supported |
| H6 | IME | → | Distance L. | 0.803 | 0.128 | 6.154 | *** | Supported |
| H7 | IME | → | Mobile L. | 0.721 | 0.123 | 6.097 | *** | Supported |
| H8 | IME | → | Self L. | 0.833 | 0.155 | 6.547 | *** | Supported |
| H9 | IME | → | Specialty L. | 0.765 | 0.123 | 6.245 | *** | Supported |

Note: IME = improving evaluation; IMC = improving communication; IMSC = improving scientific; SOL = satisfaction of learning; L. = learning; S.E. = standard error; CR = composite reliability; RA = reliability analysis. *p* = *p*-value, H = hypothesis.

Moreover, Figure 4 revealed the R2 value ranges from 0 to 1, with higher values indicating a greater explanatory power. As a guideline, R2 values of 0.75, 0.50, and 0.25 are considered to be significant, moderate, and weak, respectively [49,53]. The values of R2 of the two models are between 0.515 and 0.716. As a result, the model has medium predictive power.

## 5. Discussion

The findings of this study have validated that the extra trees regression algorithm is an effective tool for the development of PSA to determine students' achievements in secondary school. This is because all tested hypotheses were found to support the students' achievements in secondary school in terms of choice of specialty, IMC, IMSC, SOL, personalized L, distance L, mobile L, self L, and specialty L, following the use of PSA.

The findings showed that PSA on mobile apps is an effective way to enhance student achievements via the e-learning revolution in the classroom. Learning is an ongoing activity. e-learning has taken center stage because self-learning is just a click away with the help of mobile apps [17]. Students may learn at their own decision and take their time using a PSA because it is feature-oriented and created by the extra trees regression algorithm.

The PSA affects the specialty choice among students, which was found to have a significant effect on the IMC, IMSC, SOL, personalized L, distance L, mobile L, and self L. This was supported by IME results, which were significantly and positively connected with specialty learning. This showed that student preference is where the decision-making process starts when it comes to specialization. Specialization is the result of a complex interrelation of student GPA expectations, branch area anticipation, and competition for open positions [32,46]. A PSA for tracking students' specialty decisions is regarded as an educational intervention found in the current study, particularly in personalized learning and distance learning. The system encourages specialty selection while also improving secondary school achievement, particularly among those who have recently chosen their specializations and those who are about to do so. With the PSA as a specialty guide, students can be supported to exhibit their abilities in relation to their achievements.

IME positively affects IMC. This revealed that the PSA guides the students in secondary schools to improve their communication with their peers and teachers regarding the instructional materials, questions, and areas of interest in specialties. Communication varies for students aiming for technique-oriented specialties [50], which might be in response to the question of how important it is that a specialty matches with the personality, interests, and talents of a student. Through chat, forums, and other features that complement student dialogue, mobile apps can also be used to encourage communication among students [18]. Collaboration between students and teachers on e-learning-aided platforms, such as a PSA, is crucial to the achievement of students [14]. Mobile applications are being employed in various facets of education, and their use has been growing quickly [54].

Moreover, IME positively affects IMSC. This finding is strongly supported by the high scientific (49%) response rate compared to literary (41%) and industry (10%), as obtained

in this study. A key objective of IMSC on mobile apps is to foster a genuine interest in science among secondary school students, which is a crucial component of scientific literacy. According to recent studies [55,56], school scientific content has been successful in achieving this IME among students, which in turn influences students' interest in science. These studies support our findings. Given the amount of research demonstrating the beneficial effects of evaluation skills on a range of learning outcomes, it is reasonable to suggest that young students have strong evaluation skills that can support them to make the right choice of specialty because they are less likely or even unable to engage with significant societal issues related to science.

Additionally, IME also positively affects the SOL. The results demonstrate that encouraging students to engage in self-evaluation of their interest areas using mobile learning apps (i.e., performance scale apps) increases their satisfaction. The obtained results were further supported by [57], who asserted that application usage affects mobile user satisfaction. The results are in line with previous studies [9,58] that have demonstrated that students are more likely to be encouraged to use PSA as an e-learning support if they find them relevant to their learning outcomes and specialty support. Students are more likely to be satisfied with a PSA when they have IME and if they find it suitable for their learning.

IME positively affects personalized L. This indicates that personalized learning enabled by PSA can promote student achievement through the school support system. Schools may need to make a variety of curricular and structural modifications as a result of personalized learning. However, the results of personalized learning point to the potential for this form of teaching to have a significant positive impact on student progress [26,59]. According to research, mobile apps can increase student involvement in school and academic success [60]. Student choice affects personalized learning, which showed a link between SOL, IMSC, achievement, and specialty choice. Researchers believe that having a choice gives students a higher sense of autonomy and competence, which in turn boosts their achievement [61].

IME positively affects distance L. This significant finding might be ascribed to the positive utilization of e-learning. The primary tool for adopting e-learning in education is the Internet and application-enabled systems. The popularity of online learning has encouraged students to assume more responsibility for their information acquisition [62]. With e-learning apps, such as the PSA, students can learn and make their own decisions on their areas of interest and specialization at secondary level. Higher GPA students have been shown to perform better academically when learning from a distance at their own pace, according to [63]. Their strong academic achievement has primarily occurred through personalized learning. They might be less comfortable in an online setting where it might be more challenging to hold in-depth conversations without a mobile app.

Similarly, IME positively affects mobile L. This finding indicates that students' academic performance might be impacted by mobile learning if they use these gadgets for academic learning. With PSA, it was discovered that mobile learning had a very large impact on students' academic achievement and that their opinions toward using their phones for learning were moderately positive in their chosen specialty. The authors of [28] studied how teachers perceived students' academic performance when mobile devices were used in the classroom and found that 91 percent of the educators at various grade levels polled believed that mobile devices have a significant impact on students' learning in the classroom. This report is consistent with our findings. There has been an increasing amount of research conducted to determine ways to help students utilize mobile devices in and outside of the classroom. However, a consensus on the most effective method has not yet been reached. With the findings of this study, performance scale applications using machine learning algorithms can be reliable mobile apps that can support secondary school students in improving their academic achievement.

Likewise, IME positively affects self L. This suggests that grades earned by students who engage in self-learning can raise their GPA relative to grades earned by students whose grades do not count toward the final grade. In support of the finding of this study, studies have indicated that students are intelligent. With mobile apps, they can be encouraged

to better self-evaluation if they are aware that their actions will affect their final choice of specialty [64–66].

### 5.1. Conclusions and Recommendations

This study has found a positive and significant effect of PSA on students' learning achievement in a secondary school in Jordan. This validates that the ML algorithm or extra trees regression is an effective tool for the development of PSA. Through the PSA, all tested hypotheses were found to support the students' academic achievements in secondary school. The findings support each of our assumptions. The effectiveness of the system developed was able to accurately predict the students' choice of specialty, IMC, IMSC, SOL, personalized L, distance L, mobile L, self L, and specialty L. The results show that students who utilized the PSA indicated positive self-learning as measured by the IME, IMC, IMSC, and SOL. Conclusively, the PSA can efficiently predict the choice of specialty and academic achievements of students in secondary schools. Therefore, this can be adopted as an e-learning tool in secondary education in Jordan.

This study recommended that future research should look into additional functionalities of the PSA, according to the findings of this study. These functionalities include response period, changes in the screen, demand per second, use of a network, usage of memory, how long it takes to complete a task, and throughput. These improvements can be conducted by future studies. Additionally, future studies could look into testing this PSA in multiple selected secondary schools on an important variable and the ability to choose multiple specialties. The variables of this study can also be considered.

### 5.2. Limitations

The results cannot be generalized because they are limited to only secondary schools in Jordan and those with comparable conditions and characteristics. The use of a single school to draw a conclusion is one of the main drawbacks. The presence of a group of students who do not want to study in general or make any effort is one of the most significant obstacles encountered in this research. Before conducting more studies in various contexts, larger samples can be considered to prove the effectiveness and viability of the application since the sample used in this study is relatively fair.

**Author Contributions:** M.A. and A.A. designed and carried out the study and contributed to the analysis of the results and to the writing of the manuscript. F.O. and D.K. designed and carried out the study, collected data, and contributed to the writing of the manuscript. All authors have read and agreed to the published version of the manuscript.

**Funding:** The authors received no financial support for the research, authorship, and/or publication of this article.

**Data Availability Statement:** The data presented in this study are available on request from the corresponding author.

**Acknowledgments:** The participation of teachers in this research is highly appreciated.

**Conflicts of Interest:** The authors declare that the research was conducted in the absence of any commercial or financial relationships that could be construed as potential conflict of interest.

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
