# Peer review of "Development and Testing of Performance Scale Application as an Effective Electronic Tool to Enhance Students’ Academic Achievements"

_electronics, doi:10.3390/electronics11234023_

Round 1

Reviewer 1 Report

The title of this article is Development of Performance Scale Apps but it focuses more on testing. In this article, it is not explained what PSA was developed, the method of developing it, the features that were superior, and the novelties that were raised. This article discusses the use of PSA, whereas the reader wants to know more about how PSA is created, the Extra Trees Regression logarithm used and the method used for ML.

in lines 351 - 392, does not show the function of Extra Trees Regression in PSA development.

PSA trials through several methods, have shown the expected results, although here only positive results are seen, while the weaknesses that appear are not shown.

This article needs to discuss more about the PSA development process, the extra trees regression method used, and then tested as dominantly shown in this article.

Author Response

Reviewer Comments

Response

Reviewer 1

The title of this article is Development of Performance Scale Apps but it focuses more on testing.

In this article, it is not explained what PSA was developed, the method of developing it, the features that were superior, and the novelties that were raised. This article discusses the use of PSA, whereas the reader wants to know more about how PSA is created, the Extra Trees Regression logarithm used and the method used for ML.

The title has been revised to reflect the content of the manuscript. This is because the research covered both the development and the testing.

The revised manuscript contains detailed information about PSA, the method used, its characteristics, and its contribution.

The Extra Trees Regression algorithm used, and the method used for ML have been included in the revised file.

Thank you

in lines 351 - 392, does not show the function of Extra Trees Regression in PSA development.

It has been revised.

PSA trials through several methods, have shown the expected results, although here only positive results are seen, while the weaknesses that appear are not shown.

Yes, because there are none. However, we encountered some challenges, as mentioned in the manuscript, but these challenges were not at analytical result level.

This article needs to discuss more about the PSA development process, the extra trees regression method used, and then tested as dominantly shown in this article.

The PSA development process, as well as the extra trees regression method that were used and tested, have been included in the revised file.

Reviewer 2 Report

The study under consideration seems interesting, however, my comments are as follows:

·         As the authors have developed a single application, so in the current title, the word Apps shall be replaced with Application.

·         The keywords

Achievement, Prediction, Secondary School, Specialty seems very general, try giving some more specific keywords, if any.

·         Only 74 students are considered in this study and the sample size seems small. I recommend increase the sample size by considering more variables response period, changes in the screen, demand per second, use of a network, usage of memory, how long it takes to complete a task, and throughput.

·         Instead of writing the syntax in lines 351-392, try to give the algorithm.

Author Response

Reviewer Comments

Response

The study under consideration seems interesting, however, my comments are as follows:

·         As the authors have developed a single application, so in the current title, the word Apps shall be replaced with Application.

·         The keywords

Thank you

It has been corrected accordingly.

Achievement, Prediction, Secondary School, Specialty seems very general, try giving some more specific keywords, if any.

These were the primary variables and focus for the PSA. These variables were chosen within the context of identified problems and needs. Currently, there are no other specific keywords other than these. Thank you for your suggestion.

·         Only 74 students are considered in this study and the sample size seems small. I recommend increase the sample size by considering more variables response period, changes in the screen, demand per second, use of a network, usage of memory, how long it takes to complete a task, and throughput.

According to the app testing literature, average of 60 participants was considered an adequate sample size by many reports. The 74 participants used in this study were considered a sufficient sample size.

However, more variables have been considered and included in the revised manuscript, as suggested. Thank you for your scholarly suggestions.

·         Instead of writing the syntax in lines 351-392, try to give the algorithm.

It has been revised as suggested.

Reviewer 3 Report

The methodology and the results of the research are clearly presented. The manuscript is very well structured and contributes to literature

1. What is the main question addressed by the research? The effect of PSA on students' learning achievement in secondary school in Jordan.

2. Do you consider the topic original or relevant in the field? Does it address a specific gap in the field? More relevant than original. It addresses future research into additional functionalities of the PSA.

3. What does it add to the subject area compared with other published material? The manuscript highlights the need of additional research to effect of PSA on students' learning achievement.

4. What specific improvements should the authors consider regarding the methodology? What further controls should be considered? The methodology is well presented. The only drawback, which is mentioned in the paragraph of ‘Limitations’ is that the results of the research cannot be generalized. However, the manuscript contributes to the literature of Jordan.

5. Are the conclusions consistent with the evidence and arguments presented and do they address the main question posed? Yes.

6. Are the references appropriate? Yes, they are and appropriate documented in the manuscript.

7. Please include any additional comments on the tables and figures. The tables present part of the statistical analysis and the figures are informative. They enhance both the testing of hypotheses and the conclusion of the research.

Author Response

Reviewer Comments

Response

The methodology and the results of the research are clearly presented. The manuscript is very well structured and contributes to literature.

Thank you, we appreciate your time.

1. What is the main question addressed by the research? The effect of PSA on students' learning achievement in secondary school in Jordan.

2. Do you consider the topic original or relevant in the field? Does it address a specific gap in the field? More relevant than original. It addresses future research into additional functionalities of the PSA.

Thank you, we appreciate your time.

3. What does it add to the subject area compared with other published material? The manuscript highlights the need of additional research to effect of PSA on students' learning achievement.

4. What specific improvements should the authors consider regarding the methodology? What further controls should be considered? The methodology is well presented. The only drawback, which is mentioned in the paragraph of ‘Limitations’ is that the results of the research cannot be generalized. However, the manuscript contributes to the literature of Jordan.

Thank you, we appreciate your time.

5. Are the conclusions consistent with the evidence and arguments presented and do they address the main question posed? Yes.

6. Are the references appropriate? Yes, they are and appropriate documented in the manuscript.

7. Please include any additional comments on the tables and figures. The tables present part of the statistical analysis and the figures are informative. They enhance both the testing of hypotheses and the conclusion of the research.

Thank you.

Reviewer 4 Report

On Table 10, "Note = 545" is overlapped by "H2".

Author Response

Reviewer Comments

Response

On Table 10, "Note = 545" is overlapped by "H2".

The overlap has been corrected. Thank you

Round 2

Reviewer 1 Report

the article has been revised and can be recommended for publication

Author Response

Response to Reviewer Comments

Reviewer Comments

Response

Reviewer 1

The article has been revised and can be recommended for publication

We appreciate your time and scholarly input.

Thank you

Reviewer 2 Report

Though the article has been improved much. However, the content of the section 2 can be improved by using the following article, please

Yousafzai, B.K.; Khan, S.A.; Rahman, T.; Khan, I.; Ullah, I.; Ur Rehman, A.; Baz, M.; Hamam, H.; Cheikhrouhou, O. Student-Performulator: Student Academic Performance Using Hybrid Deep Neural Network. Sustainability 202113, 9775. https://doi.org/10.3390/su13179775

Author Response

Response to Reviewer Comments

Reviewer Comments

Response

Reviewer 2

Though the article has been improved much. However, the content of the section 2 can be improved by using the following article, please

Yousafzai, B.K.; Khan, S.A.; Rahman, T.; Khan, I.; Ullah, I.; Ur Rehman, A.; Baz, M.; Hamam, H.; Cheikhrouhou, O. Student-Performulator: Student Academic Performance Using Hybrid Deep Neural Network. Sustainability 202113, 9775. https://doi.org/10.3390/su13179775

The manuscript has been improve as suggested using the suggested article. 

We appreciate your time and scholarly input.

Thank you
